# Brief communication: Preliminary ICESat-2 measurements of outlet glaciers reveal heterogeneous patterns of seasonal dynamic thickness change

Christian J. Taubenberger[1,2], Denis Felikson[1,3], Thomas Neumann[1]

[1]Cryospheric Sciences Laboratory, NASA Goddard Space Flight Center, Greenbelt, MD, 20771, United States of America

[2]Environmental Health and Engineering Dept., Johns Hopkins University, Baltimore, MD, 21218, United States of America

[3]Goddard Earth Sciences Technology and Research Studies and Investigations II, Morgan State University, Baltimore, MD 21251, United States of America

*Correspondence to*: Christian J. Taubenberger (ctaubenberger@gmail.com)

**Abstract**

Dynamic changes of marine-terminating outlet glaciers are projected to be responsible for about half of future ice loss from the Greenland Ice Sheet. However, we lack a unified, process-based understanding that can explain the observed dynamic changes of all outlet glaciers. Many glaciers undergo seasonal dynamic thickness changes and classifying the patterns of seasonal thickness change can improve our understanding of the processes that drive glacier behavior. The Ice, Cloud and land Elevation Satellite (ICESat-2) provides space-based, seasonally repeating altimetry measurements of the ice sheets, allowing us to quantify near-termini seasonal dynamic thickness patterns of 37 outlet glaciers around the Greenland Ice Sheet. We classify the glaciers into seven common patterns of seasonal thickness change over a two-year period from 2019 to 2020. We find small groupings of neighboring glaciers with similar seasonal thickness change patterns but, within larger sectors of the ice sheet, seasonal thickness change patterns are mostly heterogeneous. Future studies can build upon our results by extending these time series, comparing seasonal dynamic thickness changes with external forcings, such as ocean temperature and meltwater runoff, and with other dynamic variables such as seasonal glacier velocity and terminus position changes.

## 1 Introduction

Understanding the complex nature of Earth's ice sheets is of critical importance as they have undergone dynamic changes in recent decades (Church et al., 2013; Oppenheimer et al., 2019). Greenland Ice Sheet (GrIS) marine-terminating outlet glaciers, which drive dynamic ice mass change, are projected to account for $50 \pm 20\%$ of total mass loss over the $21^{st}$ century (Choi et al., 2021). While multi-year and decadal changes of ice sheet discharge via outlet glaciers have been studied before (Mouginot et al., 2019), patterns of seasonal thickness change have not yet been studied for a representative sample of GrIS outlet glaciers. Outlet glaciers exhibit seasonal fluctuations in velocity with distinct patterns (Moon et al., 2014; Vijay et al.,

2019; Vijay et al., 2021) but the lack of seasonal thickness change measurements contributes to a lack of understanding of what processes control glacier dynamics on seasonal time scales. Seasonal thickness changes of outlet glaciers are driven by both external forcings (e.g., precipitation, evaporation, runoff, terminus melt) and internal glacier dynamics (e.g., subglacial and englacial hydrology, terminus calving) and classifying their patterns of seasonal thickness change is the first step towards a more holistic understanding of the processes that control them. Prior work has used satellite altimetry to study seasonal

surface elevation changes of the ice sheet (e.g., Johannessen et al., 2005; McMillian et al., 2016; Sutterley et al., 2018; Gray et al., 2019). Here, we focus on measuring dynamic ice sheet thickness changes near the termini of 37 GrIS outlet glaciers at seasonal resolution using the ATL06 land ice along-track altimetry dataset from the Ice, Cloud and land Elevation Satellite-2 (ICESat-2; Markus et al, 2017; Neumann et al, 2019). Large scale observational studies such as this allow for smaller, less studied, glaciers to be observed at the same time as more well-studied glaciers and comparisons to be drawn into how these

lesser-known glaciers compare with the seasonal thinning of larger glaciers, which is critical for better understanding the drivers of dynamic change in a changing climate across all outlet glaciers. We use each glacier's temporal pattern of seasonal dynamic thickness changes to group glaciers into 7 distinct patterns over 2019 and 2020. We use the spatial distribution of glacier patterns to investigate whether they can be attributed to atmospheric forcing, with the hypothesis that glaciers exhibit similar seasonal patterns within regions on the order of several hundreds of kilometers, commensurate with mesoscale

atmospheric circulation patterns. Given that we present just one to two years of data, our results are not intended to definitively characterize these glaciers but, rather, to present a method for quantifying seasonal dynamic thickness changes and to highlight the heterogeneity exhibited by these glaciers over the study time period. We discuss ways in which future work could build on our results in Section 4.

## 2     Data and methods

We used three data sources within this study: (1) The ATLAS/ICESat-2 L3A Land Ice Height, Version 3 (ATL06) data product, acquired by the Advanced Topographic Laser Altimeter System (ATLAS) instrument on board the ICESat-2 observatory, which provides geolocated measurements of land-ice surface heights (Smith et al., 2019); (2) Making Earth System Data Records for Use in Research Environments (MEaSUREs) glacier termini dataset of annual Greenland outlet glacier locations from Synthetic Aperture Radar (SAR) mosaics and Landsat 8 OLI imagery, version 1 (Joughin et al., 2015), from which we

use outlet glacier locations and identifier (ID) numbers; (3) Arctic Digital Elevation Model Mosaic (ArcticDEM; Porter et al, 2018), a digital surface elevation model of the GrIS that we used as a reference height dataset to remove along- and across-track surface slopes from the ATL06 measurements.

    ATL06 provides measurements of ice sheet surface elevation at an along-track spatial resolution of 20 m, which allows for ample spatial sampling of the fast-flowing, dynamic portions of GrIS outlet glaciers (Smith et al., 2020). We use elevation

data (h_li) retrieved from all six ATLAS ground tracks to achieve the highest density of data available. ICESat-2 has a repeat cycle of 91 days, allowing for sufficient temporal sampling to measure seasonal changes of glaciers, although we do not receive

data from every satellite pass due to cloud interference. We filter out poor quality ATL06 height data using the ATL06 quality summary flag (atl06_quality_summary), keeping only data for which the flag is set to zero.

The MEaSUREs glacier termini dataset contains locations for 238 glaciers across the GrIS, as well as an ID number (Joughin et al, 2015). We selected 65 glaciers from the MEaSUREs dataset due to their spatial distribution across several GrIS regions and range of average ice velocities between 68 m/yr and 8141 m/yr (Rignot and Mouginot, 2012). The 65 glaciers chosen for this study also correspond to the glaciers for which a dense record of terminus positions has been generated by the Calving Front Machine (CALFIN; Cheng, 2020). The CALFIN dataset is currently the only pan-Greenland dataset of seasonal terminus positions. Although we do not use this dataset in this study, due to the fact that currently available CALFIN data does not extend past mid-2019, our selection of glaciers will enable comparisons of seasonal thickness change with seasonal terminus position in future studies. We define glacier seasons by three-month periods of winter (Dec-Jan-Feb), spring (Mar-Apr-May), summer (Jun-Jul-Aug), and autumn (Sep-Oct-Nov). We removed glaciers that do not contain a full year (4 seasons) of ICESat-2 data from either 2019 or 2020, reducing the number of glaciers categorized to 42 (listed in supplementary spreadsheet).

To collect ATL06 measurements representative of near-terminus glacier thickness change, we created a 2 km x 2 km bounding box for each glacier, centered on each glacier's location in the MEaSUREs dataset, within which we aggregated ATL06 data. We manually adjusted the MEaSUREs glacier locations slightly to ensure between one and three ICESat-2 repeat ground tracks intersect each box but we kept each bounding box within 10 km of the terminus for each glacier. The 4km$^2$ bounding box was chosen as an arbitrary size, however it was kept to this size as a larger box may include data off the main fast flowing section of the outlet glacier.

The ArcticDEM Mosaic represents the mean ice sheet surface elevation between ~2015 and 2016 (Porter et al., 2018). The DEM has a 32-m spatial resolution and is used as the reference ice sheet surface elevation to account for the surface slope of the glaciers. Because the repeating passes of ICESat-2 do not exactly survey the same location on the surface of the ice sheet (particularly in the first 9 months of the ICESat-2 mission), ATL06 measurements from season to season are affected by both the vertical component of surface elevation change as well as differences in surface elevation due to surface slope. To account for this, we sampled the ArcticDEM at each ATL06 measurement and subtracted the ArcticDEM elevation from each ATL06 surface elevation measurement. This effectively changes the datum of the ATL06 measurements to the ArcticDEM, thereby accounting for the surface slope of the ice sheet within our bounding boxes, leaving just the vertical component of surface elevation differences.

We use the ATL06 data within each bounding box, a surface mass balance model, and a firn model to calculate each glacier's dynamic thickness change from season to season. For each glacier, we calculate the surface elevation change (dH) between ICESat-2 observations and the ArcticDEM. We then calculated the seasonal dynamic dH as the mean of the dHs within each bounding box for each year and season, and we subtracted the surface elevation change due to changes in surface mass balance (SMB) and firn air content changes using output from the Community Firn Model (CFM; Medley et al., 2020), forced by Modern-Era Retrospective analysis for Research and Applications, Version 2 (MERRA-2) climate reanalysis (Gelaro

et al., 2017). Over the two-year timescale of our study, we assumed constant bed elevation and, thus, our surface elevation change measurements are equal to ice thickness change. We removed the trend from each glacier's seasonal dynamic dH, calculated over the entire duration of the available data to isolate the seasonal fluctuations from the longer-term trend. We removed 5 of the 42 glaciers with measurements of seasonal dynamic dH larger than 50 m over one season, assuming that these are errors (Joughin et al., 2020), leaving 37 glaciers for which we classified seasonal dynamic dH patterns.

To account for uncertainty in seasonal dynamic dH, we propagated error through our calculations from each data source with the assumption of random, uncorrelated error. We used the error estimates provided by ATL06 to account for error on each height data point (h_sigma). We conservatively assume 5 m of random error in the ArcticDEM elevations, although the actual uncertainty in ArcticDEM elevations is likely less than this value (Noh and Howat, 2015). We assume a 20% uncertainty on the thickness change due to SMB and firn components, estimated by the CFM. Assuming uncorrelated and random errors in the ATL06 and ArcticDEM surface elevation measurements, we used standard error propagation rules to calculate the error on seasonal dynamic dH, $\sigma_{s.d.dH}$:

$$\text{Equation 1: } \sigma_{s.d.dH} = \frac{1}{n}\left(\sum_{i=1}^{n} \sigma_{h\_li,i}^2 + 5^2\right)^{1/2} + 0.2 \times |dH_{CFM}|$$

where $\sigma_{h\_li,i}$ represents the error on each ATL06 surface elevation measurement (h_li_sigma), 5 m represents the error in each ArcticDEM surface elevation, $n$ represents the number of ATL06 elevations within the bounding box for a particular season, and $dH_{CFM}$ is the absolute value of the magnitude of surface elevation change due to changes in SMB and firn air content changes from CFM. We do not account for uncertainty in the trend that is removed from each glacier's seasonal dynamic dH because the trend is removed solely to present the thickness changes more clearly in plots. Quantifying uncertainty in the dynamic thickness change trend could be done more thoroughly in future studies, given more ICESat-2 data that will be collected over the coming years. Additionally, keeping the trend in the seasonal dynamic dH has no impact on our categorization of glacier behavior for all but five glaciers, as we discuss in Section 4.

Using the time series of seasonal dynamic dH for each glacier, we manually grouped glaciers into categories based on their seasonal patterns of thickness change. Because seasonal dynamic dH had not been surveyed for a representative set of GrIS outlet glaciers, we did not prescribe categories prior to generating results. Instead, we based the categories on the timing of observed seasonal dynamic thinning and thickening for our surveyed glaciers. These classifications are based on the difference from one season to the next, rather than at each point in time. Each year of data is individually categorized; in other words, the classification for one glacier in 2019 does not influence the classification of the same glacier in 2020.

## 3 Results

We find that, over 2019 and 2020, the 37 surveyed glaciers can be categorized into seven seasonal patterns: no statistically significant seasonal change, mid-year thinning, mid-year thickening, winter-to-spring and summer-to-autumn thinning with spring-to-summer thickening, spring-to-summer thinning with winter-to-spring and summer-to-autumn thickening, sharp

single season thickening, and full-year thickening (Fig. 1). Glaciers were classified as "no statistical seasonal change" if seasonal dynamic dH uncertainties were larger than the amplitude of seasonal change across all seasons within a given year. Sharp single season thickening includes glaciers that undergo a lone season of significant (>3 times the change between any

other seasons and >3 times the uncertainties for that glacier) thickening (either spring or summer) followed immediately by a similar sharp decline in thickness. Rink Isbrae is the best example of this, undergoing 6-10 m of change during this spike (Fig. 1E). Mid-year thickening refers to glaciers exhibiting two consecutive seasons of thickening from winter-to-spring and spring-to-summer before thinning from summer-to-autumn. Conversely, mid-year thinning glaciers exhibit winter-to-spring and spring-to-summer thinning with thickening from summer-to-autumn. Each glacier's detrended dynamic thickness change,

alongside the seasonal trend of SMB and total dH change is plotted in the supplementary materials (Figs. S1 through S34). Although we have removed the trend to better illustrate seasonal dynamic dH for each glacier, we note that keeping the trend in the data alters our classifications for just five of the surveyed glaciers: Alanngorliup Sermia (Fig. S2), Kangerlussuup Sermia (Fig. S16), Kakivfaat Sermiat (Fig. S27), Cornell Gletsjer (Fig. S32), and Nansen Gletsjer (Fig. S47). Without the trend removed from the dynamic dH, there is a thinning trend in 2019 for Kangerlussuup Sermia (Fig. S16) and Kakivfaat Sermiat

(Fig. S27), across both years for Cornell Gletsjer (Fig. S32), and in 2020 for Nansen Gletsjer (Fig. S47). Alanngorliup Sermia (Fig. S2) exhibits a slight overall thickening. These glaciers exhibit strong one-to-two-year trends and although, for example, there is little seasonal change over 2019 for Kangerlussuup Sermia in their detrended seasonal dynamic dHs, the glacier is actually thinning overall across throughout the year without annual trend removed. What this does highlight, is that for all other glaciers, their seasonal dynamic thickness changes are larger in magnitude than changes due to the 1- or 2-year trend

and, thus, our classification is not sensitive to the removal of the trend. That being said, in general, care must be taken when interpreting seasonal changes with a trend removed that has been estimated from just 1 or 2 years of data.

We find that the 37 surveyed GrIS outlet glaciers are well distributed across the seven patterns. Figure 2 shows glacier classifications for both 2019 and 2020 in the table but displays the classification from the earliest available year on the map. With each year individually categorized, there are 51 total seasonal cycles observed between 2019 (30) and 2020 (21). Of these

seasonal cycles, there are 15 seasonal cycles exhibit spring-to-summer thickening with winter-to-spring and summer-to-fall thinning, 13 seasonal cycles experience mid-year thinning, 9 seasonal cycles within the spring-to-summer thinning and winter-to-spring and summer-to-fall thickening pattern, 7 seasonal cycles with mid-year thickening, 2 seasonal cycles with sharp single season thickening, 1 seasonal cycle exhibiting full-year thickening, and 4 seasonal cycles with no statistical seasonal change pattern. Of the 14 glaciers for which we have two years of data, we find that most glaciers exhibit seasonal thickness

change patterns that differ from year to year. Two glaciers exhibit repeating patterns: Ussing Braer N (Fig. S31) and Alison Gletsjer (Fig. S35). However, the remaining glaciers, for which ICESat-2 can so far provide two annual cycles worth of data, exhibit changing patterns between 2019 and 2020.

Although there are spatial clusters of glaciers with similar seasonal thickness change patterns, there is heterogeneity within the regions that contain multiple surveyed glaciers (Fig. 2). We use the 2019 classifications, for all glaciers with data in 2019,

to compare glaciers per region because we have more glaciers classified in that year (30 glaciers) than in 2020. In the NW, 6

glaciers exhibit a mid-year thinning pattern, 5 glaciers exhibit spring-to-summer thinning with winter-to-spring and summer-to-fall thickening, 2 exhibit spring-to-summer thickening with winter-to-spring and summer-to-fall thinning, 2 exhibit mid-year thickening, 1 glacier exhibits sharp single season thickening, and 2 exhibit no statistically significant change. In the CW, 3 glaciers exhibit spring-to-summer thinning with winter-to-spring and summer-to-fall thickening, 3 glaciers exhibit mid-year thinning, 2 glaciers exhibit mid-year thickening, 1 glacier exhibits sharp single season thickening, and 1 glaciers exhibit no statistically significant change. Within the SE, 6 glaciers exhibit spring-to-summer thickening with winter-to-spring and summer-to-fall thinning, and 1 glacier exhibits a mid-year thinning pattern. In the N, the single surveyed glacier, Petermann Gletsjer, exhibits spring-to-summer thickening with winter-to-spring and summer-to-fall thinning in 2019, but switches to mid-year thickening in 2020. Small clusters of neighboring glaciers with similar patterns can be seen in the NW with some form of mid-year or summer thinning (glacier IDs 31, 32, 34, and 35), the CW (glacier IDs 5, 7, 8, and 9), and the SE presents the most homogeneity, with 6 glaciers exhibiting the same pattern (glacier IDs 147, 148, 153, 158, 169, and 173) but there is no one pattern that is representative of all glaciers within each region.

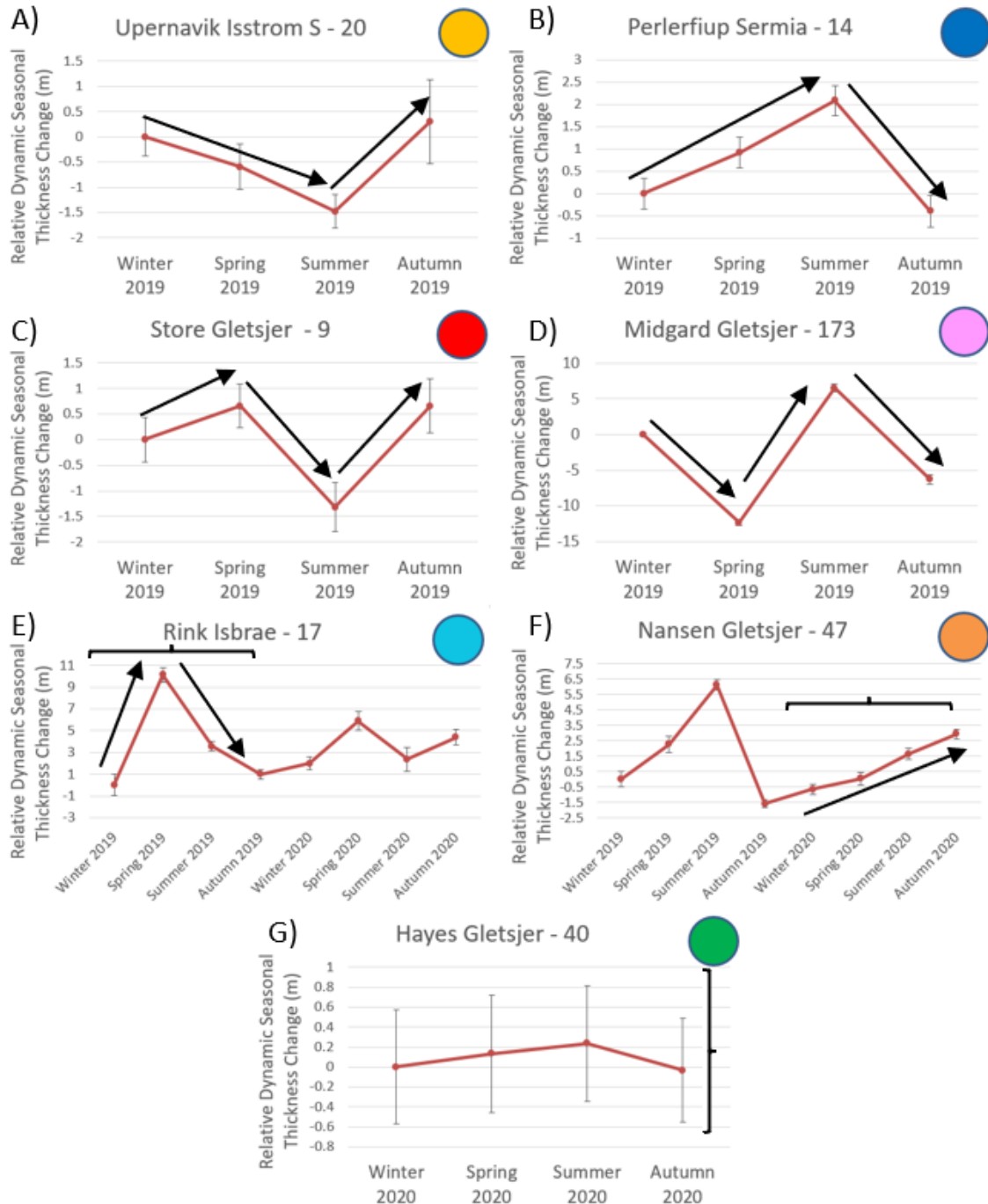

**Figure 1.** Patterns of outlet glacier dynamic seasonal thickness change with annual trend removed: A) mid-year thinning, B) mid-year thickening, C) summer thinning with spring and autumn thickening, D) spring and autumn thinning with summer thickening, E) sharp single season thickening, F) full-year thickening, and G) no statistically significant change. Curly brackets highlight the full-year thickening pattern of Nansen Gletsjer in 2020 (F) and the extent of the error bars encompassing no seasonal change for Hayes Gletsjer (G). Each value plotted is relative to the first value in the time series, which is shifted to zero.

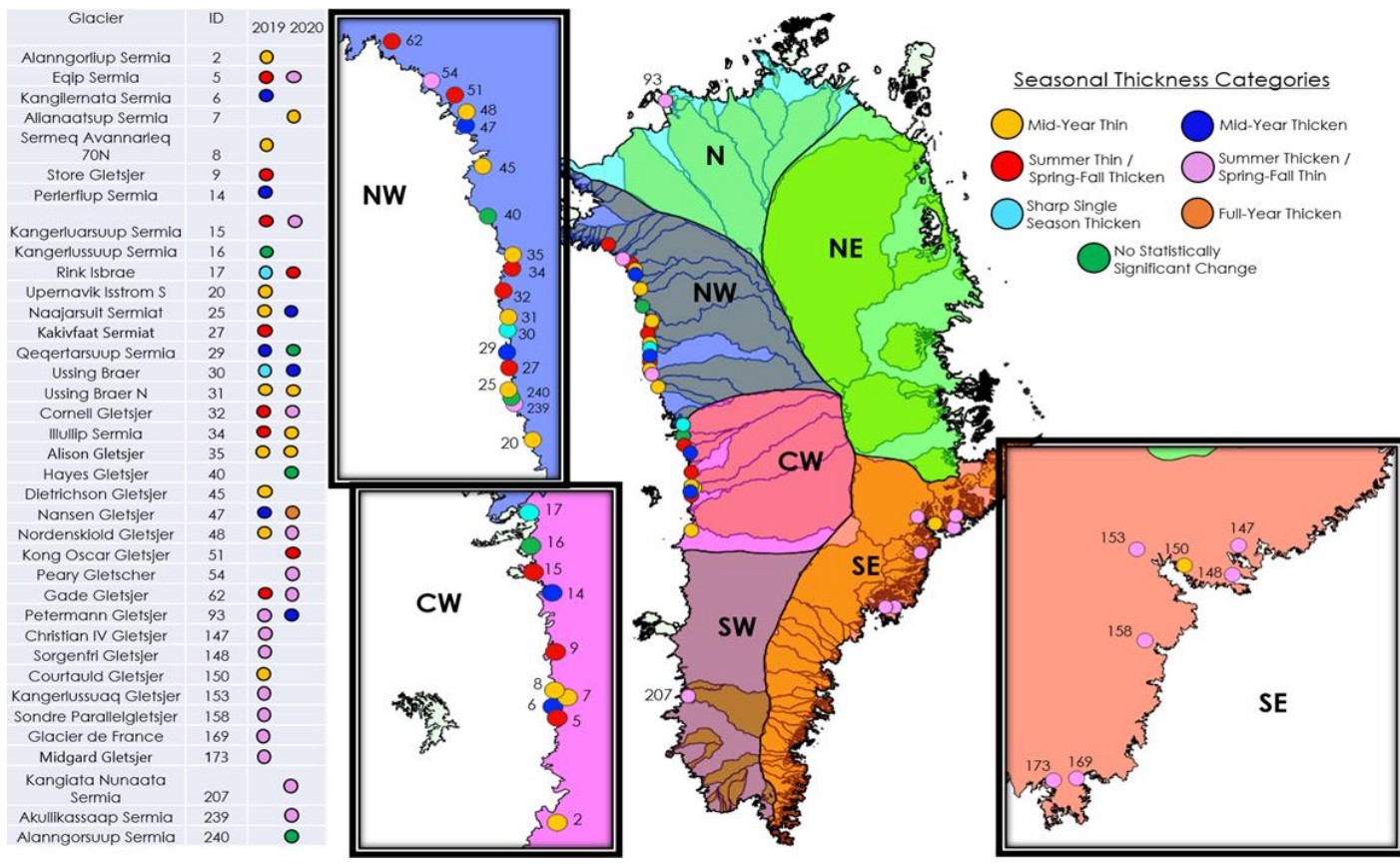

**Figure 2.** Locations, seasonal dynamic thickness change patterns, and average ice speeds of 37 GrIS outlet glaciers. Glaciers with different patterns in 2019 and 2020 are depicted on the ice sheet map with their 2019 pattern coloration, while both 2019 and 2020 patterns are shown in yearly pattern left-side table.

## 4 Discussion and conclusions

Enabled by 91-day repeat measurements from ICESat-2, we have developed the first classification of GrIS outlet glacier seasonal dynamic thickness change patterns for a representative sample of glaciers from around the ice sheet. We have chosen to use the ATL06 data product and to account for along- and across-track surface slopes using the ArcticDEM as a reference elevation dataset. This method allowed us to aggregate surface elevation data within customized bounding boxes, representative of each glacier's behavior. Higher-level data products, such as ATL11 and ATL15, will provide estimates of surface elevation change through time and we believe it will be worthwhile for future work to compare our results against the higher-level ICESat-2 products, both to build confidence in our results and as a check on the data products themselves.

Our results reveal little regional coherency in seasonal dynamic thickness change patterns, outside of the southeast region, indicating that mesoscale atmospheric circulation patterns are not the likely driver of differences in patterns among glaciers. While we do find small clusters of similar patterns, we do not observe similar patterns across the larger north-west or central-west ice sheet regions. If atmospheric forcing (or errors in our model for the atmospheric forcing) were the primary driver of seasonal dynamic thickness changes, we would expect to see coherent patterns of seasonal changes across each region. However, we do not find this to be the case, indicating that other factors that differ from glacier to glacier within each region are causing the differences in observed patterns. This finding is consistent with seasonal glacier velocity changes, which also exhibit spatial heterogeneity (Moon et al., 2014; Vijay et al. 2019; Vijay et al., 2021). Ocean forcing may be responsible for the differences in seasonal dynamic thickness change patterns because heat transport from the continental shelf to the termini of outlet glaciers is modulated by fjord geometry, which is heterogeneous among glaciers (Carroll et al., 2017). Each glacier's unique geometry, including both fjord geometry and subglacial bed topography, which have been shown to govern observed differences in terminus retreat (Catania et al., 2018), and the multi-annual upstream diffusion of thinning (Felikson et al., 2021), may also be responsible for the observed heterogeneity in seasonal thickness changes. Additionally, glacier geometry may influence each glacier's dynamic seasonal response by modulating the effects of changes in driving stress and surface melt, driven by atmospheric forcing.

Refining the ATL06 data quality flag (atl06_quality_summary), with the goal of accepting additional good-quality measurements that are currently flagged as poor-quality, would benefit future studies of seasonal outlet glacier change by increasing the data volume available. Because ICESat-2 has a repeat cycle of 91 days, collecting good-quality data from each pass is critical to studies of the seasonal thickness changes of outlet glaciers. The current set of parameters used by the ATL06 quality summary flag may remove good-quality measurements over rough topography, high surface slopes, or low-reflectivity surfaces under clouds (Smith et al., 2021). In the course of our study, we found that 12 additional glaciers, of the subset of 65 glaciers we initially selected from the MEaSUREs dataset, could be included in our results, had we ignored the quality summary flag entirely. Of course, some of the measurements that are removed by the quality summary flag are unusable and we do not advocate ignoring data quality checks entirely. However, we suggest that further inspection of the parameters used for the quality summary flag to potentially reduce the strictness by which data is eliminated may prove useful and would allow additional glaciers to be considered in future ICESat-2 data releases.

As ICESat-2 continues data collection, future work should build on our two-year assessment of seasonal dynamic thickness changes by extending our record and comparing with other glacier variables and external forcings. The MEaSUREs dataset identifies 239 total outlet glaciers around the ice sheet and, by adding more outlet glaciers and extending the record forward in time, future studies can examine how consistent the patterns are from year to year, identify new patterns not exhibited by the glaciers in our study, and better identify glaciers that exhibit the same or different patterns through time. With a longer and more comprehensive classification of seasonal thickness changes, future work can focus on compiling a holistic record of seasonal glacier dynamics by investigating thickness changes together with terminus position and velocity changes. The subset of glaciers that we have selected for study are ones that have a temporally rich dataset of terminus position changes

from the newly developed CALFIN automated deep learning extraction method (Cheng et al., 2020) as well as from added sources in the recent TermPicks (Goliber et al., 2021) dataset, which will allow our results to be directly compared with seasonal terminus positions once CALFIN data is extended into late 2019 and 2020. Finally, to advance our understanding of the processes that drive seasonal glacier behavior, future work should compare seasonal dynamic thickness changes with external forcings such as seasonal ocean temperature changes and surface meltwater runoff estimates. Our study provides the first classification of seasonal dynamic thickness changes of outlet glaciers around the GrIS to complement previous classifications of seasonal velocity change (Moon et al., 2014; Vijay et al. 2019; Vijay et al., 2021), bringing us one step closer to a holistic understanding of seasonal glacier dynamics.

**Author contribution.** C.T. and D.F. conceptualized the experiment and goals. C.T. carried out the experiment, developed the code, and performed the simulations. C.T. prepared the manuscript with written review and editing from D.F and T.N. T.N. performed project administration and funding acquisition.

**Data and code availability.** The Supplementary Information associated with this brief communication contains the seasonal thickness change measurements presented in the manuscript, along with the surface mass balance component of seasonal thickness change from the Community Firn Model and MERRA-2. Additionally, a shapefile of locations of the glaciers surveyed is provided.

**Competing interests.** There are no competing interests to disclose about this brief communication.

**Acknowledgements.** This work was performed through the NASA Goddard Space Flight Center Internship Program, administered by the Goddard Space Flight Center Office of Education and funded by the ICESat-2 Project Science Office. Resources supporting this work were provided by the NASA High-End Computing (HEC) Program through the NASA Center for Climate Simulation (NCCS) at the Goddard Space Flight Center. We thank Brooke Medley for providing output from the Community Firn Model.

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
