# Peer review of "Brief communication: Preliminary ICESat-2 measurements of outlet glaciers reveal heterogeneous patterns of seasonal dynamic thickness change"

_The Cryosphere, 2021_

## Referee Comment (RC2)

Review of Brief Communication: **ICESat-2 reveals seasonal thickness change patterns of Greenland Ice Sheet outlet glaciers for the first time**, Taubenberger et al.

This study uses two years of ICESat-2 elevation data to categorize 34 outlet glaciers based on their patterns of seasonal thickness change near the terminus. The classification is based on patterns of detrended dynamic thickness change, which is isolated from the total elevation change using modeled SMB and firn compaction with a reference GIMP DEM. The authors discuss seven main types of seasonal thickness change, show that pattern distribution is spatially heterogeneous, and conclude that the fastest glaciers in their sample undergo spring/summer thickening. The manuscript presents a valuable assessment of seasonal glacier change that leverages the spatial and temporal resolution of new ICESat-2 elevation data and is therefore timely and relevant to ongoing research in the community. The manuscript is well written with a very nice Figure 2 and includes an extensive (and helpful) Supplement. That said, there are several issues with how seasonal change is quantified and presented/discussed (primarily the loss of seasonal signal from detrending a short time series and the limit this method presents for seasonal interpretations) that prevent this manuscript from being more valuable. These changes, described below, do not require assimilating additional data or extensive reanalysis, and can probably be done fairly quickly.

**Main comments:**

**Detrending data:**
I wonder if it is useful or appropriate to detrend such short time series, especially at glaciers with only one year of data. This approach can completely alter the characteristics of the seasonal pattern and/or the magnitude of change in a given season (see for example, thinning from Autumn 2019 to Spring 2020 at Nansen glacier in the raw time series as opposed to thickening shown over the same period in the detrended time series).

**Timing and description of thickness change:**
I think the timing of seasonal thickness change can be presented in a way that is more intuitive. Based on Figure 1 and the supporting text, a given seasonal change is defined by the change from the previous season to the current (for example, an increase in thickness from spring to summer is described as "summer thickening" – as in Figure 1c). However, this description could be misleading to readers. For example, an elevation increase between mid-March (spring) to mid-June (summer) would be considered summer thickening in the text, even though the time over which the change occurred is primarily spring months (April and May).

Instead, I suggest one of two alternatives:
   (1) I think it would be more accurate to describe the timing of thickness change by the season that corresponds to the midpoint between two observations. For example, thickening observed between mid-June and mid-September would be centered on early August, or "summer thickening" (as opposed to autumn thickening due to a September end point).

(2) The second alternative is to change the language surrounding seasonal change throughout the manuscript. Rather than describing a pattern as one with "summer thickening," describe the glacier as one that "thickened from spring to summer."

Relatedly, the vales plotted in Figure 1 are somewhat challenging to interpret. My understanding is that dH is first derived by comparing ICESat-2 values to those from GIMP DEM, and then the dH time series is shifted so that the first data point (typically in winter) has a value of zero (see the purple dynamic time series in the Supplement). These time series are then detrended, with some examples representative of each seasonal pattern appearing in Figure 1. Due to the detrending, the resulting series then start with a positive or negative wintertime value that is somewhat meaningless. Without carefully reading the methods and cross-referencing the Supplementary figures, I would incorrectly interpret a negative wintertime value to represent wintertime dynamic thinning. In this case, the values themselves are unimportant, but rather it is the *change* between seasonal values that has meaning. I would suggest shifting the time series to begin at zero post-detrending and changing the y-axis label to read "Relative dynamic thickness change". If the authors elect to forego detrending altogether as I suggest above, a sentence can be added to the figure caption here to describe how values are relative to the initial time series surface. Alternatively, a secondary y-axis could be added to each figure that shows the derivative of thickness change values, or the change between each season.

**On excluding seasonal changes greater than 25 meters:**
Some justification for the 25m-magnitude seasonal change cutoff, which is used to exclude several glaciers from the analysis, is warranted. Sub-annual thickness changes of similar magnitude have been discussed in the literature (for example, up to 50 m at Jakobshavn in Joughin et al., 2020 and 19 m at Helheim in Bevan et al., 2015).

Joughin, I., E. Shean, D., E. Smith, B. and Floricioiu, D.: A decade of variability on Jakobshavn Isbræ: Ocean temperatures pace speed through influence on mélange rigidity, Cryosphere, 14(1), 211–227, doi:10.5194/tc-14-211-2020, 2020.

Bevan, S. L., Luckman, A., Khan, S. A. and Murray, T.: Seasonal dynamic thinning at Helheim Glacier, Earth Planet. Sci. Lett., 415, 47–53, doi:10.1016/j.epsl.2015.01.031, 2015.

**Glacier speed vs seasonal dynamic change pattern, beginning on line 160**
Velocity values are taken straight from Rignot and Mouginot 2012, which is cited, but these velocity figures are now ~1 decade older than the ICESat-2 elevation data used in this study. Was a regression performed to conclude a weak relationship mentioned in line 160, or rather a more qualitative assessment? These things considered, I'm not sure this paragraph/analysis adds much to the study as is probably best omitted.

If velocity and dynamic thickness changes are more closely examined in future work, I'd suggest using the seasonal range in glacier speed (perhaps as a % of annual mean speed) as a more appropriate metric to compare against dynamic thickness change patterns. For example, I would hypothesize that glaciers with larger seasonal ranges in velocity to have seasonal thickness patterns that are less sensitive to SMB, due to a larger dynamic thickness change component.

**Minor comments:**

**Line 63**
Change "average ice discharges" to "average ice velocities"

**Courtrauld**, a slow-moving glacier, is reported as having one of the largest total dynamic thickness changes of ~20m. Do the authors have a suggestion for why this could be possible?

This leads me to a related point: while fully incorporating terminus change data from CALFIN might be outside the scope of this paper, it would be useful to note how significant front change plays a role at glaciers with the largest observed change. For example, Courtauld (#150) is a relatively small glacier – did it undergo a rapid retreat and readvance to account for such a large ~20 m dynamic thinning over the 2-year observational period noted above?

**On Figure 2**
Glacier 34 is classified as having summer thinning (defined in the manuscript as having a decrease between spring and summer in the detrended time series) in 2019, but this is not supported by the figure in the Supplement, which shows thickening in the detrended time series.

**Line 191**: "Our results reveal little regional coherency in seasonal dynamic thickness change patterns, indicating that atmospheric circulation patterns are not the likely driver of differences in patterns among glaciers"
Was this a hypothesis in the study? What are the mechanisms that could potentially link atmospheric circulation to the dynamic thickness change (SMB removed) over such short time scales?

Consider adding a citation to the recent paper with updated velocity classifications to the intro:

Vijay, S., King, M., Howat, I., Solgaard, A., Khan, S., & Noël, B. (2021). Greenland ice-sheet wide glacier classification based on two distinct seasonal ice velocity behaviors. Journal of Glaciology, 1-8. doi:10.1017/jog.2021.89

---

## Author Comment (AC2)

**Author Response to RC1**

**General Response:** We would like to thank the reviewer for their thoughtful and constructive comments. Here, we provide our responses to all comments, and we explain the revisions that we plan to make to the manuscript. One change that we plan to make that we would like to highlight, in response to comments from Reviewer 2, is that we are going to remove discussion of average glacier speeds. Instead, as suggested in the comments below, if the data is available, we will add a comparison of seasonal dynamic thickness changes to terminus position changes for seven glaciers, each one being representative of a particular seasonal dynamic thickness change pattern. We leave a more comprehensive comparison to future studies, once there is more data collected by ICESat-2 that can provide seasonal dynamic thickness change patterns for more outlet glaciers around Greenland.

**Major Points:**

1. My primary concern with the presented analysis is the use of the GIMP DEM as the reference surface. If I follow the methods correctly, the authors subtract the GIMP DEM from the ICESat-2 elevations so that the ICESat-2 elevations are effectively converted to anomalies and slope effects are removed. Why use the GIMP DEM which, as the authors state, represents the mean ice sheet surface from 2003-2009? The ice sheet has evolved considerably since that time and the ArcticDEM should be more accurate and closer in time to the ICESat-2 observations. Thus, if the ArcticDEM is used as a reference, the vertical offsets due to imprecise repeats over a sloping surface should be more accurately removed from the analysis.

Author Response: We agree with this change and we have changed our reference DEM from GIMP to ArcticDEM. The reviewer is correct in that this change will allow for a better representation of the current ice sheet, and the error on each seasonal dynamic thickness change measurement in the plots (Figure 1, Supplementary Material Figures) decreased for many glaciers. Using ArcticDEM also allowed for the analysis of three new glaciers, Kakivfaat Sermiat (27), Alison Gletsjer (35), and Midgard Gletsjer (173) because the dynamic seasonal thickness change for these glaciers now falls within the 50m threshold that we used to discard bad data (this threshold was previously 25m but has been increased, based on a comment from Reviewer 2).

2. I appreciate the transparency in the process by which the glaciers were selected, but I find it curious that the glaciers were selected in part based on their inclusion in the CALFIN detailed terminus position dataset yet these data were not included in the analysis. Why were the CALFIN data not included in the analysis? The authors state that the inclusion of terminus position time series in such an analysis would be beneficial and it seems as though those data are available, but simply not included here. I do not think a detailed inter-comparison is necessary but it would be helpful to know if seasonal patterns in terminus position and thickness are correlated. The preliminary analysis could focus on centerline terminus change and could aggregate the changes across all glaciers to determine if there is any hint of a relationship between the variables. A more detailed analysis could then be presented in another paper.

Author Response: As stated by the reviewer, a comprehensive comparison between terminus changes and thickness changes would be a better fit in a future study. Part of our reasoning for not including a comparison between our results and the CALFIN dataset is that there is limited overlap between ICESat-2 seasonal thickness change patterns and CALFIN, which currently provides data through mid-2019. In the future, as additional ICESat-2 is available to characterize seasonal thickness change patterns for more outlet glaciers around the ice sheet, a more in-depth study could be conducted.

However, we agree that a qualitative discussion of seasonal centerline terminus changes would fit well within this paper. To address this, we will add time series of the centerline position change for one glacier from each of our characterized patterns to qualitatively investigate the relationship. Using these data, we will discuss the magnitude and timing of terminus change in comparison to dynamic thickness change. Currently, we are investigating new data from a recent paper (Goliber, et al., 2021) released after we had submitted our manuscript. While this TermPicks dataset, which combines the CALFIN data with many other sources of terminus positions, also does not provide sufficient temporal overlap with our results to compare with our seasonal thickness changes, it is our hope that the creators of CALFIN or TermPicks can provide us with additional data to extend the terminus position time series past mid-2019.

Reference:
Goliber, S., Black, T., Catania, G., Lea, J. M., Olsen, H., Cheng, D., Bevan, S., Bjørk, A., Bunce, C., Brough, S., Carr, J. R., Cowton, T., Gardner, A., Fahrner, D., Hill, E., Joughin, I., Korsgaard, N., Luckman, A., Moon, T., Murray, T., Sole, A., Wood, M., and Zhang, E.: TermPicks: A century of Greenland glacier terminus data for use in machine learning applications, The Cryosphere Discuss. [preprint], https://doi.org/10.5194/tc-2021-311, in review, 2021.

Minor Points:

- line 63: Are the units on discharge correct? Normally discharge refers to mass or volume per unit time, not a unit of length per unit time.

Author Response: We will replace "discharge" with "velocity". This was a typo.

- line 84: Instead of "vertical component of surface elevation change", I recommend "vertical component of surface elevation differences" since the word change has the connotation of differences over time and the removal of slope effects is meant to isolate the vertical component of the full difference (both due to spatial offsets and temporal changes) in surface elevation observations.

Author Response: We will make this change.

- line 94: Add a space between numbers and units ("25 m").

Author Response: We will make this change.

- lines 118-122: No statistical seasonal change is the first in the long list of categories and is also listed in the following sentence.

Author Response: We will change the wording to ensure this is clearer and avoid repetition.

- lines 160-171: I recommend renaming "medium-fast speed" and "medium-slow" to "moderately fast" and "moderately slow".

Author Response: We will make this change.

- lines 196-202: It is worth noting in this section that the variability can be driven by atmospheric forcing even if the variability is unlikely to be directly driven by variations in surface mass balance. The authors point out that the geometry of fjords may be incredibly important in regard to the access of warm waters to glacier termini, but the underlying topography of the glacier may also influence the dynamic response of the glacier to changes in meltwater fluxes and/or driving stresses driven by atmospheric change.

Author Response: We agree with the reviewer and we will clarify the statement that, alongside fjord geometry, subglacial bed topography also plays an important role in how glaciers respond to atmospheric forcing (via changes to surface melt and driving stress).

---

## Author Comment (AC3)

**Author Response to RC2**

**General author response**: We would like to thank the reviewer for their thoughtful and constructive comments. Here, we provide our responses to all comments, and we explain the revisions that we plan to make to the manuscript. Along with the specific changes outlined in the responses below, we have also made a change from using the GIMP DEM to using the ArcticDEM as our reference ice sheet surface, in response to a comment made by Reviewer 1. This allows for a better representation of the current ice sheet, as the reference elevations are closer in time with the ICESat-2 observations, and has significantly reduced many glaciers' individual mean dH error values for each point seen in the plots (Figure 1, Supplementary Material Figures).

**Main comments:**
Detrending data:
I wonder if it is useful or appropriate to detrend such short time series, especially at glaciers with only one year of data. This approach can completely alter the characteristics of the seasonal pattern and/or the magnitude of change in a given season (see for example, thinning from Autumn 2019 to Spring 2020 at Nansen glacier in the raw time series as opposed to thickening shown over the same period in the detrended time series).

Author Response: We agree with the reviewer that care should be taken when using a short time series to estimate a trend. What we have found, however, is that the impact of removing the trend is very limited for this dataset (more information below). Because our goal is to classify seasonal change, we have chosen to keep the detrended measurements as the focus in the main text. However, we will add a paragraph to the discussion section to discuss the implications of removing the trend and the impact on our classifications.

Part of our reasoning for keeping the detrended measurements as the focus of the paper is that the impact of detrending the data on our seasonality classifications is limited to just 5 out of 37 glaciers when using ArcticDEM (new plots will be provided in the revised manuscript). In other words, 5 glaciers would change classifications if we were to use the dynamic thickness change time series without the trend removed (i.e., comparing the purple and orange curves in the supplementary figures for glaciers 2, 16, 27, 32, and 47). This is also true for the classifications in the initial submission of the paper, which used the GIMP DEM as the reference ice sheet surface height. Four glaciers (IDs 16, 47, 93, 150) would have changed classifications if we had used the dynamic thickness change without the trend removed (see supplementary figures in our first submission). Nevertheless, we will add to our discussion the caveat that the interpretation of seasonal changes of these glaciers is sensitive to the way in which a trend is estimated and removed. The fact that our classification of most glaciers is unaffected by the trend shows that the dynamic thickness changes that these glaciers undergo from season to season are larger in magnitude than their annual trend and we will add this as a point of discussion, as well.

Timing and description of thickness change:
I think the timing of seasonal thickness change can be presented in a way that is more intuitive. Based on Figure 1 and the supporting text, a given seasonal change is defined by the change from the previous season to the current (for example, an increase in thickness from spring to summer is described as "summer thickening" – as in Figure 1c). However, this description could be misleading to readers. For example, an elevation increase between mid-March (spring) to mid-June (summer) would be considered summer thickening in the text, even though the time over which the change occurred is primarily spring months (April and May).

Instead, I suggest one of two alternatives:
(1) I think it would be more accurate to describe the timing of thickness change by the season that corresponds to the midpoint between two observations. For example, thickening observed between mid-June and mid-September would be centered on early August, or "summer thickening" (as opposed to autumn thickening due to a September end point).

(2) The second alternative is to change the language surrounding seasonal change throughout the manuscript. Rather than describing a pattern as one with "summer thickening," describe the glacier as one that "thickened from spring to summer."

Author Response: We agree with the comment that the language we used to describe the timing of change may be ambiguous and we will change the language using the reviewer's second suggested alternative throughout the manuscript. Each point in our plots in Figure 1 shows the mean surface elevation change, as referenced to the initial point. Thus, seasonal surface elevation change is the difference between one point and the next. We will make it clearer that our interpretation and classifications are based on the difference from season to season, rather than at each point individually.

Relatedly, the vales plotted in Figure 1 are somewhat challenging to interpret. My understanding is that dH is first derived by comparing ICESat-2 values to those from GIMP DEM, and then the dH time series is shifted so that the first data point (typically in winter) has a value of zero (see the purple dynamic time series in the Supplement). These time series are then detrended, with some examples representative of each seasonal pattern appearing in Figure 1. Due to the detrending, the resulting series then start with a positive or negative wintertime value that is somewhat meaningless. Without carefully reading the methods and cross-referencing the Supplementary figures, I would incorrectly interpret a negative wintertime value to represent wintertime dynamic thinning. In this case, the values themselves are unimportant, but rather it is the change between seasonal values that has meaning. I would suggest shifting the time series to begin at zero post-detrending and changing the y-axis label to read "Relative dynamic thickness change". If the authors elect to forego detrending altogether as I suggest above, a sentence can be added to the figure caption here to describe how values are relative to the initial time series surface. Alternatively, a secondary y-axis could be added to each figure that shows the derivative of thickness change values, or the change between each season.

Author Response: We agree that the way in which we plotted the data may be unclear and, to address this comment, we will make the following changes:
1. Shift the detrended time series such that the first point is at zero
2. Change the y-axis label of our plots to "relative dynamic seasonal thickness change (m)"
3. Add a sentence to the figure caption to describe that each value is relative to the first value in the time series
4. Clarify the language in the manuscript to state that our classifications are based on the changes from season to season, rather than at each point individually

On excluding seasonal changes greater than 25 meters:
Some justification for the 25m-magnitude seasonal change cutoff, which is used to exclude several glaciers from the analysis, is warranted.
Sub-annual thickness changes of similar magnitude have been discussed in the literature (for example, up to 50 m at Jakobshavn in Joughin et al., 2020 and 19 m at Helheim in Bevan et al., 2015).

Joughin, I., E. Shean, D., E. Smith, B. and Floricioiu, D.: A decade of variability on Jakobshavn Isbræ: Ocean temperatures pace speed through influence on mélange rigidity, Cryosphere, 14(1), 211–227, doi:10.5194/tc-14-211-2020, 2020.

Bevan, S. L., Luckman, A., Khan, S. A. and Murray, T.: Seasonal dynamic thinning at Helheim Glacier, Earth Planet. Sci. Lett., 415, 47–53, doi:10.1016/j.epsl.2015.01.031, 2015.:

Author Response: We agree with the reviewer's comment and we will change our threshold to 50 m and we will cite Joughin et al. (2020) as the largest observed seasonal thickness change to date. We note, however, that this change will not add any previously excluded glaciers from our analysis because the excluded glaciers had magnitudes of dynamic thickness changes of over 75 m from season to season. Nevertheless, we will change our threshold to be consistent with previous literature.

Glacier speed vs seasonal dynamic change pattern, beginning on line 160
Velocity values are taken straight from Rignot and Mouginot 2012, which is cited, but these velocity figures are now ~1 decade older than the ICESat-2 elevation data used in this study. Was a regression performed to conclude a weak relationship mentioned in line 160, or rather a more qualitative assessment? These things considered, I'm not sure this paragraph/analysis adds much to the study as is probably best omitted.

Author Response: Please see our response to the next comment, which addresses this comment as well.

If velocity and dynamic thickness changes are more closely examined in future work, I'd suggest using the seasonal range in glacier speed (perhaps as a % of annual mean speed) as a more appropriate metric to compare against dynamic thickness change patterns. For example, I would hypothesize that glaciers with larger seasonal ranges in velocity to have seasonal thickness patterns that are less sensitive to SMB, due to a larger dynamic thickness change component.

Author Response: We agree that the qualitative assessment of the velocity values added little to the manuscript and we will remove it. Our ultimate goal, beyond the scope of this study, is to compare seasonal dynamic thickness changes with seasonal velocity changes. But we feel that this would be better left for future work that can take advantage of additional ICESat-2 measurements over the next several years, which can provide data for glaciers that we were unable to classify. In this Brief Communication, our goal is to present initial observations of thickness changes as observed by ICESat-2 as well as one potential method for quantifying these changes.

Minor comments:
Line 63
Change "average ice discharges" to "average ice velocities"

Author Response: We will make this change.

Courtrauld, a slow-moving glacier, is reported as having one of the largest total dynamic thickness changes of ~20m. Do the authors have a suggestion for why this could be possible?

Author Response: The large thickness changes we measured on Courtauld Gletsjer were due to errors in the GIMP DEM in this region. In response to a comment from Reviewer 1, we have changed our ice sheet surface elevation reference dataset to ArcticDEM. By making this switch, the errors for Courtrauld have been eliminated and we are now measuring ~2-3 m of dynamic thickness changes per season. This update will be reflected in the revisions to the manuscript.

This leads me to a related point: while fully incorporating terminus change data from CALFIN might be outside the scope of this paper, it would be useful to note how significant front change plays a role at glaciers with the largest observed change. For example, Courtauld (#150) is a relatively small glacier – did it undergo a rapid retreat and readvance to account for such a large ~20 m dynamic thinning over the 2-year observational period noted above?

Author Response: We agree with the reviewer, however, based on currently published data from the CALFIN and TermPicks (Goliber et al., 2021) datasets, glacier centerlines are only recorded until mid-2019 which does not provide enough overlap with the ICESat-2 observations. If additional CALFIN data can be provided by the creators, we will add measurements of terminus position change along glacier centerlines for one glacier from each of our seven categories. We feel that a comprehensive comparison of terminus position changes with our measurements of dynamic thickness changes for all glaciers is outside of the scope of this Brief Communication because this kind of analysis would benefit from additional years of data collected by ICESat-2. Nevertheless, if the data is available, we plan to show terminus position changes of seven glaciers, one representing each different type of seasonal pattern. This will serve as a qualitative exploration of whether seasonal thickness changes are related to terminus motion. We are in the process of investigating which seven glaciers have sufficient terminus position data in the recently released TermPicks dataset and we may or may not choose Courtauld, specifically.

Reference:
  Goliber, S., Black, T., Catania, G., Lea, J. M., Olsen, H., Cheng, D., Bevan, S., Bjørk, A., Bunce, C., Brough, S., Carr, J. R., Cowton, T., Gardner, A., Fahrner, D., Hill, E., Joughin, I., Korsgaard, N., Luckman, A., Moon, T., Murray, T., Sole, A., Wood, M., and Zhang, E.: TermPicks: A century of Greenland glacier terminus data for use in machine learning applications, The Cryosphere Discuss. [preprint], https://doi.org/10.5194/tc-2021-311, in review, 2021.

On Figure 2
Glacier 34 is classified as having summer thinning (defined in the manuscript as having a decrease between spring and summer in the detrended time series) in 2019, but this is not supported by the figure in the Supplement, which shows thickening in the detrended time series.

Author Response: The x-axis of the plot for Illullip Sermia in Supplementary Material had a mistake, where every value was shifted by one season, giving the appearance of a different categorization. This will be corrected in the new Supplementary Material, and it will correctly correspond with summer thinning.

Line 191: "Our results reveal little regional coherency in seasonal dynamic thickness change patterns, indicating that atmospheric circulation patterns are not the likely driver of differences in patterns among glaciers"
Was this a hypothesis in the study? What are the mechanisms that could potentially link atmospheric circulation to the dynamic thickness change (SMB removed) over such short time scales?

Author Response: We will add a discussion of the links between atmospheric circulation and dynamic thickness change to the introduction to better contextualize this section of the manuscript.

Consider adding a citation to the recent paper with updated velocity classifications to the intro:

Vijay, S., King, M., Howat, I., Solgaard, A., Khan, S., & Noël, B. (2021). Greenland ice-sheet wide glacier
classification based on two distinct seasonal ice velocity behaviors. Journal of Glaciology, 1-8.
doi:10.1017/jog.2021.89

Author Response: The citation will be added to the text.

---

## Author Response (AR1)

Dear Dr. Wouters,

Thank you very much for continuing to serve as editor for our manuscript despite your time constraints. We greatly appreciate your assistance in facilitating this process. We have made the requested revisions to the manuscript and figures. Given that our responses to the reviewers were formatted as point-by-point replies and there were no other specific changes
5  requested by yourself, we have simply reiterated those prior responses here in one combined document. Following that, we have also gone through and attached a marked-up version of the manuscript that contains the primary location of the requested changes by both reviewers and by Andy Shepherd highlighted as comments in the margins, as we hope that to be helpful.

Please see our specific responses to each of the comments below:

10  **Reviewer 1:**

**RC1 Major Points:**

1. My primary concern with the presented analysis is the use of the GIMP DEM as the reference surface. If I follow the methods correctly, the authors subtract the GIMP DEM from the ICESat-2 elevations so that the ICESat-2 elevations are effectively converted to anomalies and slope effects are removed. Why use the GIMP DEM which, as
15  the authors state, represents the mean ice sheet surface from 2003-2009? The ice sheet has evolved considerably since that time and the ArcticDEM should be more accurate and closer in time to the ICESat-2 observations. Thus, if the ArcticDEM is used as a reference, the vertical offsets due to imprecise repeats over a sloping surface should be more accurately removed from the analysis.

Author Response: We agree with this change and we have changed our reference DEM from GIMP to ArcticDEM. The
20  reviewer is correct in that this change will allow for a better representation of the current ice sheet, and the error on each seasonal dynamic thickness change measurement in the plots (Figure 1, Supplementary Material Figures) decreased for many glaciers. Using ArcticDEM also allowed for the analysis of three new glaciers, Kakivfaat Sermiat (27), Alison Gletsjer (35), and Midgard Gletsjer (173) because the dynamic seasonal thickness change for these glaciers now falls within the 50m threshold that we used to discard bad data (this threshold was previously 25m but has been increased, based on a comment
25  from Reviewer 2).

2. I appreciate the transparency in the process by which the glaciers were selected, but I find it curious that the glaciers were selected in part based on their inclusion in the CALFIN detailed terminus position dataset yet these data were not included in the analysis. Why were the CALFIN data not included in the analysis? The authors state that the inclusion of terminus position time series in such an analysis would be beneficial and it seems as though those data
30  are available, but simply not included here. I do not think a detailed inter-comparison is necessary but it would be helpful to know if seasonal patterns in terminus position and thickness are correlated. The preliminary analysis could focus on centerline terminus change and could aggregate the changes across all glaciers to determine if there is any hint of a relationship between the variables. A more detailed analysis could then be presented in another paper.

35  Author Response: As stated by the reviewer, a comprehensive comparison between terminus changes and thickness changes would be a better fit in a future study. Part of our reasoning for not including a comparison between our results and the CALFIN dataset is that there is limited overlap between ICESat-2 seasonal thickness change patterns and CALFIN, which currently

provides data through mid-2019. In the future, as additional ICESat-2 is available to characterize seasonal thickness change patterns for more outlet glaciers around the ice sheet, a more in-depth study could be conducted.

40

However, we have confirmed that at this time, we do not have the terminus position data required for comparison to our results. The CALFIN dataset (Cheng et al., 2021) provides terminus positions but only through mid-2019 at this point. The TermPicks dataset (Goliber et al., 2021), which includes the CALFIN data, provides additional terminus positions in 2019 and 2020 but these data are not frequent enough for us to be able to draw conclusions about the timing of glacier dynamic thickness changes at a seasonal timescale. Because of this, we will not be adding terminus position data to our manuscript and we leave this analysis to future work, when additional ICESat-2 and terminus position data will be available.

Reference:

Cheng, D., Hayes, W., Larour, E., Mohajerani, Y., Wood, M., Velicogna, I., and Rignot, E.: Calving Front Machine (CALFIN): Glacial Termini Dataset and Automated Deep Learning Extraction Method for Greenland, 1972–2019, The Cryosphere, https://doi.org/10.5194/tc-15-1663-2021, 2021.

Goliber, S., Black, T., Catania, G., Lea, J. M., Olsen, H., Cheng, D., Bevan, S., Bjørk, A., Bunce, C., Brough, S., Carr, J. R., Cowton, T., Gardner, A., Fahrner, D., Hill, E., Joughin, I., Korsgaard, N., Luckman, A., Moon, T., Murray, T., Sole, A., Wood, M., and Zhang, E.: TermPicks: A century of Greenland glacier terminus data for use in machine learning applications, The Cryosphere Discuss. [preprint], https://doi.org/10.5194/tc-2021-311, in review, 2021.

**RC1 Minor Points:**

- line 63: Are the units on discharge correct? Normally discharge refers to mass or volume per unit time, not a unit of length per unit time.

Author Response: We will replace "discharge" with "velocity". This was a typo.

- line 84: Instead of "vertical component of surface elevation change", I recommend "vertical component of surface elevation differences" since the word change has the connotation of differences over time and the removal of slope effects is meant to isolate the vertical component of the full difference (both due to spatial offsets and temporal changes) in surface elevation observations.

Author Response: We will make this change.

- line 94: Add a space between numbers and units ("25 m").

Author Response: We will make this change.

- lines 118-122: No statistical seasonal change is the first in the long list of categories and is also listed in the following sentence.

Author Response: We will change the wording to ensure this is clearer and avoid repetition.

- lines 160-171: I recommend renaming "medium-fast speed" and "medium-slow" to "moderately fast" and
70     "moderately slow".

Author Response: We will make this change.

- lines 196-202: It is worth noting in this section that the variability can be driven by atmospheric forcing even if the variability is unlikely to be directly driven by variations in surface mass balance. The authors point out that the geometry of fjords may be incredibly important in regard to the access of warm waters to glacier termini, but the
75     underlying topography of the glacier may also influence the dynamic response of the glacier to changes in meltwater fluxes and/or driving stresses driven by atmospheric change.

Author Response: We agree with the reviewer and we will clarify the statement that, alongside fjord geometry, subglacial bed topography also plays an important role in how glaciers respond to atmospheric forcing (via changes to surface melt and driving stress).

80

**Reviewer 2:**

**RC2 Main comments:**
Detrending data:
I wonder if it is useful or appropriate to detrend such short time series, especially at glaciers with
85 only one year of data. This approach can completely alter the characteristics of the seasonal
pattern and/or the magnitude of change in a given season (see for example, thinning from
Autumn 2019 to Spring 2020 at Nansen glacier in the raw time series as opposed to thickening
shown over the same period in the detrended time series).

90 Author Response: We agree with the reviewer that care should be taken when using a short time series to estimate a trend. What we have found, however, is that the impact of removing the trend is very limited for this dataset (more information below). And, because our goal is to classify seasonal change, we have chosen to keep the detrended measurements as the focus in the main text. However, we will add a paragraph to the discussion section to discuss the implications of removing the trend and the impact on our classifications.
95
Part of our reasoning for keeping the detrended measurements as the focus of the paper is that the impact of detrending the data on our seasonality classifications is limited to just 5 out of 37 glaciers when using ArcticDEM. In other words, 5 glaciers would change classifications when using the dynamic thickness change time series without the trend removed (i.e., compare the purple and orange curves in the supplementary figures for glaciers 2, 16, 27, 32, and 47). This is also true for the current
100 GIMP DEM classifications, where only a limited number (4 glaciers: IDs 16, 47, 93, 150 (data error)) would change classifications. We will add to our discussion the caveat that the interpretation of seasonal changes of these glaciers is sensitive to the way in which a trend is estimated and removed. The fact that our classification of most glaciers is unaffected by the trend shows that the dynamic thickness changes that these glaciers undergo from season to season are larger in magnitude than their annual trend and we will add this as a point of discussion as well.
105

Timing and description of thickness change:
I think the timing of seasonal thickness change can be presented in a way that is more intuitive.
Based on Figure 1 and the supporting text, a given seasonal change is defined by the change
110 from the previous season to the current (for example, an increase in thickness from spring to
summer is described as "summer thickening" – as in Figure 1c). However, this description could

be misleading to readers. For example, an elevation increase between mid-March (spring) to mid-June (summer) would be considered summer thickening in the text, even though the time over which the change occurred is primarily spring months (April and May).

Instead, I suggest one of two alternatives:
(1) I think it would be more accurate to describe the timing of thickness change by the season that corresponds to the midpoint between two observations. For example, thickening observed between mid-June and mid-September would be centered on early August, or "summer thickening" (as opposed to autumn thickening due to a September end point).

(2) The second alternative is to change the language surrounding seasonal change throughout the manuscript. Rather than describing a pattern as one with "summer thickening," describe the glacier as one that "thickened from spring to summer."

Author Response: We agree with the comment that the language we used to describe the timing of change may be ambiguous and we will change the language using the reviewer's second suggested alternative throughout the manuscript. Each point in our plots in Figure 1 shows the mean surface elevation change, as referenced to the initial point. Thus, seasonal surface elevation change is the difference between one point and the next. We will make it clearer that our interpretation and classifications are based on the difference from season to season, rather than at each point individually.

Relatedly, the vales plotted in Figure 1 are somewhat challenging to interpret. My understanding is that dH is first derived by comparing ICESat-2 values to those from GIMP DEM, and then the dH time series is shifted so that the first data point (typically in winter) has a value of zero (see the purple dynamic time series in the Supplement). These time series are then detrended, with some examples representative of each seasonal pattern appearing in Figure 1. Due to the detrending, the resulting series then start with a positive or negative wintertime value that is somewhat meaningless. Without carefully reading the methods and cross-referencing the Supplementary figures, I would incorrectly interpret a negative wintertime value to represent wintertime dynamic thinning. In this case, the values themselves are unimportant, but rather it is the change between seasonal values that has meaning. I would suggest shifting the time series to begin at zero post-detrending and changing the y-axis label to read "Relative dynamic thickness change". If the authors elect to forego detrending altogether as I suggest above, a sentence can be added to the figure caption here to describe how values are relative to the initial time series surface. Alternatively, a secondary y-axis could be added to each figure that shows the derivative of thickness change values, or the change between each season.

Author Response: We agree that the way in which we plotted the data may be unclear and, to address this comment, we will make the following changes:
1. Shift the detrended time series such that the first point is at zero
2. Change the y-axis label of our plots to "relative dynamic seasonal thickness change (m)"
3. Add a sentence to the figure caption to describe that each value is relative to the first value in the time series
4. Clarify the language in the manuscript to state that our classifications are based on the changes from season to season, rather than at each point individually

On excluding seasonal changes greater than 25 meters:
Some justification for the 25m-magnitude seasonal change cutoff, which is used to exclude several glaciers from the analysis, is warranted.
Sub-annual thickness changes of similar magnitude have been discussed in the literature (for example, up to 50 m at Jakobshavn in Joughin et al., 2020 and 19 m at Helheim in Bevan et al., 2015).

Joughin, I., E. Shean, D., E. Smith, B. and Floricioiu, D.: A decade of variability on Jakobshavn Isbræ: Ocean temperatures pace speed through influence on mélange rigidity, Cryosphere, 14(1), 211–227, doi:10.5194/tc-14-211-2020, 2020.
Bevan, S. L., Luckman, A., Khan, S. A. and Murray, T.: Seasonal dynamic thinning at Helheim Glacier, Earth Planet. Sci.
165    Lett., 415, 47–53, doi:10.1016/j.epsl.2015.01.031, 2015.:

Author Response: We agree with the reviewer's comment and we will change our threshold to 50 m and we will cite Joughin et al. (2020) as the largest observed seasonal thickness change to date. We note, however, that this change will not add any previously excluded glaciers from our analysis because the excluded glaciers had magnitudes of dynamic thickness changes
170    of over 75 m from season to season. Nevertheless, we have changed our threshold to be consistent with previous literature.

175    Glacier speed vs seasonal dynamic change pattern, beginning on line 160
Velocity values are taken straight from Rignot and Mouginot 2012, which is cited, but these velocity figures are now ~1 decade older than the ICESat-2 elevation data used in this study. Was a regression performed to conclude a weak relationship mentioned in line 160, or rather a more qualitative assessment? These things considered, I'm not sure this paragraph/analysis adds
180    much to the study as is probably best omitted.

Author Response: Please see our response to the next comment, which addresses this comment as well.

If velocity and dynamic thickness changes are more closely examined in future work, I'd suggest
185    using the seasonal range in glacier speed (perhaps as a % of annual mean speed) as a more appropriate metric to compare against dynamic thickness change patterns. For example, I would hypothesize that glaciers with larger seasonal ranges in velocity to have seasonal thickness patterns that are less sensitive to SMB, due to a larger dynamic thickness change component.

190    Author Response: We agree that the qualitative assessment of the velocity values added little to the manuscript and we will remove this from the manuscript. Our ultimate goal is to compare seasonal dynamic thickness changes with velocity changes but we feel that this is outside of the scope of this Brief Communication, in which our goal is to present initial measurements of thickness changes as observed by ICESat-2.

195

**RC2 Minor comments:**
Line 63
Change "average ice discharges" to "average ice velocities"

200    Author Response: We will make this change

Courtrauld, a slow-moving glacier, is reported as having one of the largest total dynamic thickness changes of ~20m. Do the authors have a suggestion for why this could be possible?

205    Author Response: The large thickness changes we measured on Courtauld Gletsjer were due to errors in the GIMP DEM in this region. In response to a comment from Reviewer 1, we have changed our ice sheet surface elevation reference dataset to Arctic DEM. By making this switch, the errors for Courtauld have been eliminated and we are now measuring ~2-3 m of dynamic thickness changes per season. This update will be reflected in the revisions to the manuscript.

210    This leads me to a related point: while fully incorporating terminus change data from CALFIN

might be outside the scope of this paper, it would be useful to note how significant front change plays a role at glaciers with the largest observed change. For example, Courtauld (#150) is a relatively small glacier – did it undergo a rapid retreat and readvance to account for such a large ~20 m dynamic thinning over the 2-year observational period noted above?

Author Response: We have confirmed that at this time, we do not have the terminus position data required for comparison to our results. The CALFIN dataset (Cheng et al., 2021) provides terminus positions but only through mid-2019 at this point. The TermPicks dataset (Goliber et al., 2021), which includes the CALFIN data, provides additional terminus positions in 2019 and 2020 but these data are not frequent enough for us to be able to draw conclusions about the timing of glacier dynamic thickness changes at a seasonal timescale. Because of this, we will not be adding terminus position data to our manuscript and we leave this analysis to future work, when additional ICESat-2 and terminus position data will be available.

On Figure 2
Glacier 34 is classified as having summer thinning (defined in the manuscript as having a decrease between spring and summer in the detrended time series) in 2019, but this is not supported by the figure in the Supplement, which shows thickening in the detrended time series.

Author Response: The x-axis of the plot for Illullip Sermia in Supplementary Material had a mistake, where every value was shifted by one season, giving the appearance of a different categorization. This has been corrected in the new Supplementary Material, and it once again corresponds with summer thinning.

Line 191: "Our results reveal little regional coherency in seasonal dynamic thickness change patterns, indicating that atmospheric circulation patterns are not the likely driver of differences in patterns among glaciers"
Was this a hypothesis in the study? What are the mechanisms that could potentially link atmospheric circulation to the dynamic thickness change (SMB removed) over such short time scales?

Author Response: We will add a discussion of the links between atmospheric circulation and dynamic thickness change to the introduction to better contextualize this section of the manuscript.

Consider adding a citation to the recent paper with updated velocity classifications to the intro:
Vijay, S., King, M., Howat, I., Solgaard, A., Khan, S., & Noël, B. (2021). Greenland ice-sheet wide glacier classification based on two distinct seasonal ice velocity behaviors. Journal of Glaciology, 1-8. doi:10.1017/jog.2021.89

Author Response: The citation will be added to the text.

Thank you again for your consideration and please let me know if there is anything else that you need from me. Please see the copy of the manuscript below with comments addressing the above.

Regards,

Christian Taubenberger

[revised manuscript text omitted]

Commented [CT3]: RC2 Minor Point 6: Citation with updated velocity classifications has been added throughout the paper where appropriate.

Commented [CT4]: CC1 Point 3: Language has been changed and references have been added to address examples of previous space-based seasonally repeating altimetry measurement work.

Commented [CT5]: RC2 Minor Point 5: Discussion of the link between atmospheric circulation and dynamic thickness change was expanded upon based on comments regarding Line 218 (Previous Line 191)

Commented [CT6]: RC1 Major Point 1: Data using GIMP DEM has been updated with data from ArcticDEM and this change has been implemented throughout the manuscript, within the text, figures, and in the supplementary material.

ATL06 provides measurements of ice sheet surface elevation at an along-track spatial resolution of 20 m, which allows for ample spatial sampling of the fast-flowing, dynamic portions of GrIS outlet glaciers (Smith et al., 2020). We use elevation data (h_li) retrieved from all six ATLAS ground tracks to achieve the highest  density of data available. ICESat-2 has a repeat cycle of 91 days, allowing for sufficient temporal sampling to measure seasonal changes of glaciers, although we do not receive data from every satellite pass due to cloud interference. We filter out poor quality ATL06 height data using the ATL06 quality summary flag (atl06_quality_summary), keeping only data for which the flag is set to zero.

The MEaSUREs glacier termini dataset contains locations for 238 glaciers across the GrIS, as well as an ID number (Joughin et al, 2015). We selected 65 glaciers from the MEaSUREs dataset due to their spatial distribution across several GrIS regions and range of average ice  velocities between 68 m/yr and 8141 m/yr (Rignot and Mouginot, 2012). The 65 glaciers chosen for this study also correspond to the glaciers for which a dense record of terminus positions has been generated by the Calving Front Machine (CALFIN; Cheng, 2020). The CALFIN dataset is currently the only pan-Greenland dataset of seasonal terminus positions. Although we do not use this dataset in this study, due to the fact that  currently available CALFIN data does not extend past mid-2019, our selection of glaciers will enable comparisons of seasonal thickness change with seasonal terminus position in future studies. We define glacier seasons by three-month periods of winter (Dec-Jan-Feb), spring (Mar-Apr-May), summer (Jun-Jul-Aug), and autumn (Sep-Oct-Nov). We removed glaciers that do not contain a full year (4 seasons) of ICESat-2 data from either 2019 or 2020, reducing the number of glaciers categorized to 42 (listed in supplementary spreadsheet).

To collect ATL06 measurements representative of near-terminus glacier thickness change, we created a 2 km x 2 km bounding box for each glacier, centered on each glacier's location in the MEaSUREs dataset, within which we aggregated ATL06 data. We manually adjusted the MEaSUREs glacier locations slightly to ensure between one and three ICESat-2 repeat ground tracks intersect each box but we kept each bounding box within 10 km of the terminus for each glacier. The 4 km$^2$ bounding box was chosen as an arbitrary size, however it was kept to this size as a larger box may include data off the main fast flowing section of the outlet glacier.

The Arctic DEM Mosaic  represent the mean ice sheet surface elevation between ~2015 and 2016  (Porter et al., 2018). The DEM has a 32-m spatial resolution and is used as the reference ice sheet surface elevation to account for the surface slope of the glaciers. Because the repeating passes of ICESat-2 do not exactly survey the same location on the surface of the ice sheet (particularly in the first 9 months of the ICESat-2 mission), ATL06 measurements from season to season are affected by both the vertical component of surface elevation change as well as differences in surface elevation due to surface slope. To account for this, we sampled the Arctic DEM at each ATL06 measurement and subtracted the Arctic DEM elevation from each ATL06 surface elevation measurement. This effectively changes the datum of the ATL06 measurements to the Arctic DEM, thereby accounting for the surface slope of the ice sheet within our bounding boxes, leaving just the vertical component of surface elevation differences.

**Commented [CT7]:** RC1 and RC2 Minor Point 1: Discharge changed to velocity

**Commented [CT8]:** RC1 Major Point 2 and RC2 Minor Point Regarding the addition of CALFIN data, as explained in the amended reply to RC1 and RC2, currently, CALFIN does not publicly provide enough data to make the requested additions.

**Commented [CT9]:** RC1 Minor Point 2: elevation change ha been changed to elevation differences.

We use the ATL06 data within each bounding box, a surface mass balance model, and a firn model to calculate each glacier's dynamic thickness change from season to season. For each glacier, we calculate the surface elevation change (dH) between ICESat-2 observations and the Arctic DEM. We then calculated the seasonal dynamic dH as the mean of the dHs within each bounding box for each year and season, and we subtracted the surface elevation change due to changes in surface mass balance (SMB) and firn air content changes using output from the Community Firn Model (CFM; Medley et al., 2020), forced by Modern-Era Retrospective analysis for Research and Applications, Version 2 (MERRA-2) climate reanalysis (Gelaro et al., 2017). Over the two-year timescale of our study, we assumed constant bed elevation  and, thus, our surface elevation change measurements are equal to ice thickness change. We removed the trend from each glacier's seasonal dynamic dH, calculated over the entire duration of the available data to isolate the seasonal fluctuations from the longer-term trend. We removed 5 of the 42 glaciers with measurements of seasonal dynamic dH larger than 50 m over one season , assuming that these are errors (Joughin et al., 2020), leaving 37 glaciers for which we classified seasonal dynamic dH patterns.

To account for uncertainty in seasonal dynamic dH, we propagated error through our calculations from each data source with the assumption of random, uncorrelated error. We used the error estimates provided by ATL06 to account for error on each height data point (h_sigma). We conservatively assume 5 m of random error in the ArcticDEM elevations, although the actual uncertainty in ArcticDEM elevations is likely less than this value (Noh and Howat, 2015).  We assume a 20% uncertainty on the thickness change due to SMB and firn components, estimated by the CFM. Assuming uncorrelated and random errors in the ATL06 and Arctic DEM surface elevation measurements, we used standard error propagation rules to calculate the error on seasonal dynamic dH, $\sigma_{s.d.dH}$:

$$\text{Equation 1: } \sigma_{s.d.dH} = \frac{1}{n}\left(\sum_{i=1}^{n} \sigma_{h\_li,i}^2 + \mathbf{5}^2\right)^{1/2} + 0.2 \times |dH_{CFM}|$$

where $\sigma_{h\_li,i}$ represents the error on each ATL06 surface elevation measurement (h_li_sigma), 5 m represents the error in each Arctic DEM surface elevation, $n$ represents the number of ATL06 elevations within the bounding box for a particular season, and $dH_{CFM}$ is the absolute value of the magnitude of surface elevation change due to changes in SMB and firn air content changes from CFM. We do not account for uncertainty in the trend that is removed from each glacier's seasonal dynamic dH because the trend is removed solely to present the thickness changes more clearly in plots. Quantifying uncertainty in the dynamic thickness change trend could be done more thoroughly in future studies, given more ICESat-2 data that will be collected over the coming years. Additionally, keeping the trend in the seasonal dynamic dH has no impact on our categorization of glacier behavior for all but  five glaciers, as we discuss in Section 4.

Using the time series of seasonal dynamic dH for each glacier, we manually grouped glaciers into categories based on their seasonal patterns of thickness change. Because seasonal dynamic dH had not been surveyed for a representative set of GrIS outlet glaciers, we did not prescribe categories prior to generating results. Instead, we based the categories on the timing

**Commented [CT10]:** RC2 Major Point 4: The cutoff for accepted magnitude of seasonal change in thickness has been updated from 25 m to 50 m, and Joughin et al., 2020 was cited.

**Commented [CT11]:** RC1 Minor Point 3: Space added betw numbers and units

of observed seasonal dynamic thinning and thickening for our surveyed glaciers. These classifications are based on the difference from one season to the next, rather than at each point in time. Each year of data is individually categorized; in other words, the classification for one glacier in 2019 does not influence the classification of the same glacier in 2020.

**3 Results**

We find that, over 2019 and 2020, the 34 surveyed glaciers can be categorized into seven seasonal patterns: no statistically significant seasonal change, mid-year thinning, mid-year thickening, winter-to-spring and summer-to-autumn thinning with spring-to-summer thickening, spring-to-summer thinning with winter-to-spring and summer-to-autumn thickening, sharp single season thickening, and full-year thickening (Fig. 1). Glaciers were classified as "no statistical seasonal change" if seasonal dynamic dH uncertainties were larger than the amplitude of seasonal change across all seasons within a given year. Sharp single season thickening includes glaciers that undergo a lone season of significant (>3 times the change between any other seasons and >3 times the uncertainties for that glacier) thickening (either spring or summer) followed immediately by a similar sharp decline in thickness. Rink Isbrae is the  best example of this, undergoing 6-10 m of change during this spike (Fig. 1E). Mid-year thickening refers to glaciers exhibiting two consecutive seasons of thickening  from winter-to-spring and spring-to-summer before thinning from summer-to- autumn. Conversely, mid-year thinning glaciers exhibit winter-to-spring and spring-to-summer thinning with thickening from  summer-to-autumn. Each glacier's detrended dynamic thickness change, alongside the seasonal trend of SMB and total dH change is plotted in the supplementary materials (Figs. S1 through S34). Although we have removed the trend to better illustrate seasonal dynamic dH for each glacier, we note that keeping the trend in the data  alters our classifications  for just five of the surveyed glaciers: Alanngorliup Sermia (Fig. S2), Kangerlussuup Sermia (Fig. S16), Kakivfaat Sermiat (Fig. S27), Cornell Gletsjer (Fig. S32), and Nansen Gletsjer (Fig. S47). Without the trend removed from the dynamic dH, there is a thinning trend in 2019 for Kangerlussuup Sermia (Fig. S16) and Kakivfaat Sermiat (Fig. S27), across both years for Cornell Gletsjer (Fig. S32), and in 2020 for Nansen Gletsjer (Fig. S47). Alanngorliup Sermia (Fig. S2) exhibits a slight overall thickening. These glaciers exhibit strong one-to-two-year trends and although, for example, there is little seasonal change over 2019 for Kangerlussuup Sermia in their detrended seasonal dynamic dHs, the glacier is actually thinning overall across throughout the year without annual trend removed. What this does highlight, is that for all other glaciers, their seasonal dynamic thickness changes are larger in magnitude than changes due to the 1- or 2-year trend and, thus, our classification is not sensitive to the removal of the trend. That being said, in general, care must be taken when interpreting seasonal changes with a trend removed that has been estimated from just 1 or 2 years of data.
* * *
**Commented [CT12]:** RC2 Main Point 3: Language has been changed in the manuscript, and the changes to supplementary fig[ures] have been made. The first point in the detrended time series was shifted to zero, the y-axis label has been changed, and the caption description has been altered.

**Commented [CT13]:** RC1 Minor Point 4: Language has been changed

**Commented [CT14]:** RC2 Main Point 2: Changes throughou[t] the manuscript to the timing and description of thickness change have been made based on the second alternative suggested by RC[2] to explain seasonal patterns more descriptively based on season t[o] season change.

**Commented [CT15]:** RC2 Main Point 1: Discussion regardi[ng] the detrending of data

We find that the 34 surveyed GrIS outlet glaciers are well distributed across the seven patterns. Figure 2 shows glacier classifications for both 2019 and 2020 in the table but displays the classification from the earliest available year on the map. With each year individually categorized, there are 51 total seasonal cycles observed between 2019 (30) and 2020 (21). Of these seasonal cycles, there are 15 seasonal cycles exhibit spring-to-summer thickening with winter-to-spring and summer-to-fall thinning, 13 seasonal cycles experience mid-year thinning, 9 seasonal cycles within the spring-to-summer thinning and winter-to-spring and summer-to-fall thickening pattern, 7 seasonal cycles with mid-year thickening,  2 seasonal cycles with sharp single season thickening, 1 seasonal cycle exhibiting full-year thickening, and 4 seasonal cycles with no statistical seasonal change pattern. Of the 13 glaciers for which we have two years of data, we find that most glaciers exhibit seasonal thickness change patterns that differ from year to year. Two glaciers exhibit repeating patterns:  Ussing Braer N (Fig. S31) and Alison Gletsjer (Fig. S35). However, the remaining glaciers, for which ICESat-2 can sofar provide two annual cycles worth of data, exhibit changing patterns between 2019 and 2020.

Although there are spatial clusters of glaciers with similar seasonal thickness change patterns, there is heterogeneity within the regions that contain multiple surveyed glaciers (Fig. 2). We use the 2019 classifications, for all glaciers with data in 2019, to compare glaciers per region because we have more glaciers classified in that year (30 glaciers) than in 2020. In the NW, 6 glaciers exhibit a mid-year thinning pattern, 5 glaciers exhibit spring-to-summer thinning with winter-to-spring and summer-to-fall thickening,  2 exhibit spring-to-summer thickening with winter-to-spring summer-to-fall thinning, 2 exhibit mid-year thickening, 1 glacier exhibits sharp single season thickening, and 2 exhibit no statistically significant change. In the CW, 3 glaciers exhibit spring-to-summer thinning with winter-to-spring and summer-to-fall thickening, 3 glaciers exhibit mid-year thinning, 3 glaciers exhibit mid-year thickening, 1 glacier exhibits sharp single season thickening, and 1 glaciers exhibit no statistically significant change. Within the SE, 6 glaciers exhibit spring-to-summer thickening with winter-to-spring and summer-to-fall thinning,  and 1 glacier exhibits a mid-year thinning pattern. In the N, the single surveyed glacier, Petermann Gletsjer, exhibits spring-to-summer thickening with winter-to-spring and summer-to-fall thinning in 2019, but switches to mid-year thickening in 2020. Small clusters of neighboring glaciers with similar patterns can be seen in the NW with some form of mid-year or summer thinning (glacier IDs 29, 30, 31, 32, 34, and 35), the CW (glacier IDs 5, 7, 8, and 9), and the SE presents the most homogeneity, with 6 glaciers exhibiting the same pattern (glacier IDs 147, 148, 153, 158, 169, and 173) but there is no one pattern that is representative of all glaciers within each region.

**Commented [CT16]:** RC2 Main Point 6 and 7: The qualitative assessment of the velocity values from Rignot and Mouginot, 20... have been removed from Figure 2, and the paragraph discussion velocity has been removed in entirety to improve manuscript flow

**Commented [CT17]:** RC1 Minor Point 5: Instead of renamin... this entire paragraph has been removed based on other recommendations.

Isstrøm S (2.5 km/yr), undergo patterns of summer or mid-year thinning. Medium-slow glaciers between 1.6 and 1.9 km/yr, such as Kangiata Nunaata Sermia (1.9 km/yr), Hayes Gletsjer (1.9 km/yr), Christian IV Gletsjer (1.8 km/yr), and Kangerlussuup Sermia (1.6 km/yr), undergo mid-year thickening. Slower glaciers, with speeds below 1.6 km/yr, are more divergent in their seasonal thickness responses, for instance Cornell Gletsjer (0.5 km/yr), Sorgenfri Gletsjer (0.3 km/yr), Sondre Parallelgletsjer (0.3 km/yr), and Courtauld Gletsjer (0.3 km/yr) are of similar speeds yet exhibit different patterns of seasonal thickness change. The slowest glacier we observe is Alangorliup Sermia (0.07 km/yr), which exhibits no statistical seasonal change in dynamic thickness.

**Commented [CT18]:** RC2 Minor Point 2: Thickness changes measured for Courtauld Gletsjer have been updated in the supplementary material using ArcticDEM, which has fixed the er seen involving the slow moving glacier's dynamic thickness chan

[Figure]

**Figure 1.** Patterns of outlet glacier dynamic seasonal thickness change with annual trend removed: A) mid-year thinning, B) mid-year thickening, C) summer thinning with spring and autumn thickening, D) spring and autumn thinning with summer thickening, E) sharp single season thickening, F) full-year thickening, and G) no statistically significant change. Curly brackets highlight the full-year thickening pattern of Nansen  Gletsjer in 2020 (F) and the extent of the error bars encompassing no seasonal change for Hayes Gletsjer (G). Each value plotted is relative to the first value in the time series, which is shifted to zero.

460

[Figure]

**Figure 2.** Locations, seasonal dynamic thickness change patterns, and average ice speeds of 34 GrIS outlet glaciers. Glaciers with different patterns in 2019 and 2020 are depicted on the ice sheet map with their 2019 pattern coloration, while both 2019 and 2020 patterns are shown in yearly pattern left-side table.

**Commented [CT19]:** Figure 2 has been remade using ArcticDEM classifications.

RC2 Minor Point 4: Classification of glacier 34 has been fixed to consistent between Figure 2 and Supplementary Material. Issue with the display of the axis in supplementary material plot of glacier 34.

[revised manuscript text omitted]

---

## Author Response (AR2)

Dear Dr. Wouters,

Thank you for your assistance during this process. We have made the requested revisions based on the comments from Reviewer 2. Please see the point-by-point below:

1.  In the revision, the newly added paragraph on page 5, after "Although we have removed the trend to better illustrate seasonal dynamic dH for each glacier,
    we note that keeping the trend in the data...." lists the five glaciers and their corresponding Supplementary figure.

    Note that the following text refers to Supp figure numbers that are the same as the glacier ID numbers, which do not appear to be correct in the revision (for example, Alanngsorliup Sermia is glacier ID 2, but is Figure S1, not Fig S2).

Author Response: We have made this change and thank the reviewer for catching this detail.

2.  In the second-to-last new sentence on page 5, I would clarify with the addition of "in at least one season":
    "What this does highlight, is that for all other glaciers, their seasonal dynamic thickness changes during at least one season are larger in magnitude than changes due to the 1- or 2-year trend and, thus, our classification is not sensitive to the removal of the trend."

Author Response: We have made this change.